# Analyzing intelligent tourism development and public services based on a fuzzy genetic hybrid system to promote environmental and cultural values

**Jinxia Lou**◉*

Hospitality Management Department, Tourism college of Zhejiang, Hangzhou, Zhejiang, China

* jinxia_lou@outlook.com

## Abstract

Environmental, cultural, and public service-dependent factors encourage the development of a country's tourism. In recent years, automated tourism development using statistical and accumulated data has been exploited to recommend attractive tourist features. This article thus discloses an intelligent development assessment method (IDAM) using cumulative factors (CFs) for deriving development-focused improvement in tourism. This method accounts for public services and environmental and cultural factors that promote tourism for better assessment. The fuzzy process identifies the maximum possible impacting factors by independently evaluating the reviewed values. Based on the reviewed values, the manipulation of factor relationships is derived to identify even trivial factors impacting development. The fuzzy outputs are thus integrated with optimistically impacting development factors to provide attractive recommendations. Such recommendations are analyzed using fuzzy data for previous and current development factors for new decisions. The system's efficiency was evaluated using the recommendation ratio, ensuring a 48.58% success rate, a development rate of 0.105%, a 4-factor detection rate, and a review-based assessment rate of 55.5% for a sample size of 5,000 visitors.

## 1. Introduction

The goal of travel is to discover nature, adventures, wonders, and societies, as well as to meet new people, engage with values, and take part in customs and events, which is encouraged by the dynamic force of tourism [1]. The growth and maintenance of a tourist industry's development brings travelers to certain or unspecific locations. Furthermore, ecological durability refers to the future-focused, deliberate effort to maintain natural resources and the legacy of social culture [2]. This safeguards environmental ecosystems while promoting human health and efficient prosperity [3]. The preservation of the environment is seen in green natural landscaping, abundant biodiversity, pristine beaches, desert steppe, sociocultural values, archaeological legacy, etc. These factors represent the level of motivation of tourists and the openness of the local people to receive them [4]. Since sustainability and tourism expansion are viewed

**Data Availability Statement:** All the data are within the paper.

**Funding:** National Social Science Foundation Art General Project, "Research on the driving mechanism and improvement path of digital

cultural consumption under the guidance of
expanding domestic demand", 23BH155.

**Competing interests:** The authors have declared
that no competing interests exist.

as interrelated concepts, sustainable and environmentally friendly tourism is strongly impacted by growth in its development based on the number of visitors [5].

Public management and service provision systems are aware of culture and recognize and embrace tradition at all levels. They evaluate intercultural relations, resulting in the broadening of disparities, the recognition of culture, and the customization of solutions to meet the requirements of cultural groups [6]. Since the public delivery of services and governance are fields that are evolving slowly toward a focus on cultural sensitivity, scholars and administrators in these fields have not yet come to a clear understanding of this idea [7]. The marginalized public, community administrators, and service delivery professionals are unfamiliar with organizational cultural understanding and structural limitations [8]. This impedes the promotion of public agencies responsible for identifying cultural competency self-evaluation tools and performance measures [9]. The workforce implements structural procedures and organizational systems for service delivery, and design leaders have a significant influence [10]. The leadership creates institutional policies, processes, and delivery systems that are either poorly prepared or inadequately structured. Such delivery systems can be improved to handle the demands and issues of racial, ethnic, and cultural groupings regarding public provision [11].

One of the sectors that has grown significantly over time in terms of revenue and the production of new technologies is tourism. The facilities and transport systems in tourist areas serve as impactful factors in the development of tourism [12]. Transportation is necessary for accessing points of interest (POIs) and determining the location's allure. In addition, chain participants have limited supply and information access [13]. For example, travel packages and static routes are created, demonstrating the absence of instruments that enable real-time itinerary planning [14]. The tourist trip design problem (TTDP) involves creating a travel schedule for tourists to visit different POIs while remaining under predetermined limits related to time, money, transportation, and visitor preferences, among other factors [15]. The TTDP is essential for enhancing visitor experiences and the prosperity and caliber of tourist places. In addition, the expansion of economic advantages and the competitiveness of the tourism supply chain rely on the TTDP. Personalized tourism routes are computed using TTDP-solving algorithms, and these routes serve as the foundation for the development of applications and intelligent mobile guides (MTGs). Similarly, the chain's actors' information is arranged and structured to enhance accessibility to tourism destinations and ease decision-making, especially in unpredictable circumstances [16, 17]. The objective of this system is to analyze and facilitate the advancement of intelligent tourist development, considering both environmental and cultural values. Public services and infrastructure connected to tourism are assessed using a fuzzy genetic hybrid technique. This technology enhances decision-making processes for sustainable tourism practices by integrating fuzzy logic and genetic algorithms. Its objective is to achieve a harmonious equilibrium between economic expansion and the conservation of natural resources and cultural legacies.

## 1.1. The main aims of this system

i. To create an intelligent tourism development assessment method using external impacting factors to provide better recommendations for tourists;

ii. To use fuzzy-based independent and relationship-combined impact factor validation to determine for trivial and nontrivial recommendations; and

iii. To discuss the proposed method's performance using external data and conduct a comparative discussion using different variables.

## 2. Related works

Al Fararni et al. [18] developed an AI-powered integrated travel recommendation system using big data. The developed method does more than recommend a list of tourist spots based on visitor selections. It operates similarly to a trip planner, creating an extensive schedule with various travel options. The method increases travel, especially to the Daraa-Fillet region. Achmad et al. [19] proposed a cooperative framework for open innovation, partners, and supporting system resources. The primary goal of the method is to provide a cooperative model of open innovation with stakeholders. The framework also facilitates support systems to create a sustainable tourism sector in a fast-paced, cutthroat market. Raising functionality can have a major and beneficial effect on the tourism sector. Castro et al. [20] suggested a qualitative examination of the factors that lead to locals' support for the growth of film tourism. The method aimed to create and evaluate a conceptual model of the factors influencing locals' attitudes. The associations between the variables and locals' support for future film tourism were investigated using partial least squares-structural equation modeling. The method shows that the most important predictor is place attachment.

Wang et al. [21] introduced creativity based on cultural legacy at locations for cultural tourism. The notion of cultural inheritance-based innovation (CIBI) and the scale for measuring it were developed from a paradoxical viewpoint. The practical performance of CIBI at heritage tourism destinations (HTDs) was determined using an assessment instrument. The method strengthens and broadens the theoretical understanding of cultural innovation in tourist locations. Li et al. [22] analyzed the effects of residents' place connection and spiritual health on serial mediation. The method examines attachment and spiritual welfare as the underlying mechanisms of the impact. The process by which locals establish their opinions on tourism growth is better understood. The method provides direction on how to increase the favorable views of inhabitants. Arif et al. [23] proposed a decentralized recommender system utilizing known and unknown rating approaches for the ambient intelligence of tourism places. The method generates recommendations for choosing tourist locations as a guide for choosing scenario visualizations. The method uses decentralized technologies to implement and manage data circulation between system components. This method allows the player to select and control the tour visualization. Him et al. [24] suggested tourism's spatial expertise with MaaS (Mobility as a Service). The method demonstrates the connections between the MaaS provision of tourism locations and the desire for travel options. The method employs a mixed logit model to analyze the conjoint choices of tourism travel portfolios. The suggested approach improves the experiences of tourists.

Han et al. [25] introduced residents' support for developing environmentally friendly tourism. This is a helpful approach for forecasting locals' support for sustainable tourism development. It involves the combination of social exchange theory and bottom-up spillover theory. This approach supports the growth of sustainable tourism. Chen et al. [26] determined variables affecting rural tourists' interactions with services. This method integrates the theories of planned behavior and service escape to investigate the elements influencing experiences in rural tourism. Positive behavioral attitudes, subjective norms, and perceived behavioral controls influence the real behavior of rural tourists. This method offers a novel framework for capturing the unique characteristics of rural landscapes. Liang et al. [27] proposed tourism arrival forecasting with an enhanced fuzzy time series approach. A dual decomposition technique was created to fully extract the key elements of the tourist arrival series and reduce the data complexity. This strategy may overcome the potential shortcomings of individual decomposition approaches. The proposed approach holds great promise for predicting future traveler arrivals.

Zhao et al. [28] suggested monitoring the long-term growth of smart travel service systems, which can strengthen the management of tourist destinations and make tourism more efficient. This method evaluates the factors influencing users' willingness to use intelligent tourism service systems (ITSSs) at scenic spots. The suggested approach provides a theoretical foundation and empirical support for the efficient and sustainable growth of ITSSs. Shojatalab et al. [29] introduced a novel multiobjective optimization framework for tourist systems utilizing fuzzy data; some of the model parameters in our suggested model are fuzzy numbers. A suitable algorithm was suggested for solving the multiobjective optimization model, and a new approach based on the epsilon constraint method was proposed for this multiobjective optimization model. The method outperformed the alternative algorithms. Cankurt et al. [30] developed a model for predicting tourism demand with a stacking combination model and an adaptive fuzzy combiner. The method aims to create an ensemble model for multivariate forecasting. This approach handles many input variables by combining neural networks with adaptive neuro-fuzzy inference systems. The developed method performs better than its stand-alone base learner counterparts.

Wang et al. [31] examined the effects of tourism development and geographical heterogeneity to investigate the income gap between urban and rural areas. By utilizing a mediating effect model, the process of tourism's impacts the income gap between urban and rural areas was determined and showed that tourism appears to have less of an effect on closing the income gap in the eastern region than in the western region. The approach serves as a bridge between the growth of the tourism industry. Kong et al. [32] presented a novel real-time processing system and Internet of Things (IoT) application designed to facilitate the development of cultural tourism. Internet of Things (IoT) sensors and big data analytics were employed to monitor and manage tourism resources effectively. The system conducts real-time data processing from many sources to generate valuable insights that inform decision-making processes. Furthermore, it integrates sophisticated algorithms to optimize the allocation of resources and improve the visitor experience while fostering the preservation of cultural assets.

Ahmad, N., & Ma, X. [33] examined the correlation between tourism growth and environmental contamination in various socioeconomic brackets of nations. The study employed panel data analysis to investigate the influence of various factors, including tourist arrivals, tourism receipts, and tourism investment, on air pollution, water pollution, and $CO_2$ emissions. The results indicated a multifaceted relationship between the expansion of tourism and the deterioration of the environment, underscoring the necessity of implementing sustainable tourism laws and practices. The authors offered suggestions for mitigating the negative environmental impacts associated with the rise of tourism.

Ali S et al. [34] examined the ecological consequences of global tourism, considering the influence of policy ambiguity, renewable energy, and service sector production. Drawing upon data obtained from prominent tourist locations, the study utilized sophisticated econometric methodologies to examine the impact of these variables on carbon emissions and ecological footprints. The study's results demonstrate complex interconnections, indicating that policy uncertainty hinders the adoption of sustainable practices. Conversely, the rise of renewable energy and the service sector have the potential to alleviate environmental deterioration resulting from tourism activities. The policy recommendations put forth by the authors aim to enhance the promotion of environmental sustainability within the tourism industry.

García-Madurga et al. [35] comprehensively analyzed the use of artificial intelligence (AI) within the tourism sector. The process involved a methodical examination of available literature studies to consolidate the present status of AI implementation in different areas of the tourism industry. The research emphasized the capacity of artificial intelligence (AI)

technology to augment tourism experiences, operations, and decision-making procedures while also revealing areas of deficiency and promising avenues for future investigation.

Bulchand-Gidumal, J. [36] investigated the effects of artificial intelligence (AI) on the travel, tourism, and hospitality industries. The study examined a range of artificial intelligence (AI) applications, including chatbots, recommendation systems, and predictive analytics, emphasizing the capacity of AI to improve customer experiences and operational efficiency. It also examined the obstacles and ethical implications of implementing artificial intelligence in certain sectors.

## 3. Proposed model

### 3.1 Intelligent Development Assessment Method (IDAM) using Cumulative Factors (CFs)

The development of the proposed method was initiated with the data representation acquired from the Ningbo Bureau of Statistics in 2023. The primary aim of this system is to promote the sustainable and rational development of tourism while simultaneously ensuring the protection of environmental and cultural resources. The objective is to achieve a state of equilibrium that promotes economic expansion via tourism and the conservation of natural resources, ecological systems, and cultural legacy. The system seeks to enhance decision-making processes about public services, infrastructure, and resource allocation in the tourism industry by utilizing a fuzzy genetic hybrid approach. The proposed hybrid methodology integrates the advantages of fuzzy logic, which can effectively address uncertainties and imprecise data, with genetic algorithms, which provide efficient search and optimization techniques. The system aims to optimize the beneficial outcomes of tourism while mitigating any adverse effects on the environment and cultural values by employing a synergistic approach. The primary objective is to advance responsible tourism strategies that foster destinations' enduring sustainability, guaranteeing forthcoming generations' ability to value and partake in their natural and cultural heritage. This source provides the influencing factors with their category and observation period from a renowned tourist location. The influencing factors are numbered from F1 to F10 for 20,000 visitors annually. The representation in Fig 1 below incorporates all of the above with the type of analysis and outcome.

The external impact factors are numbered from F1 to F10 and are categorized as environmental, cultural, or public service. The assessments are validated using fuzzy and genetic

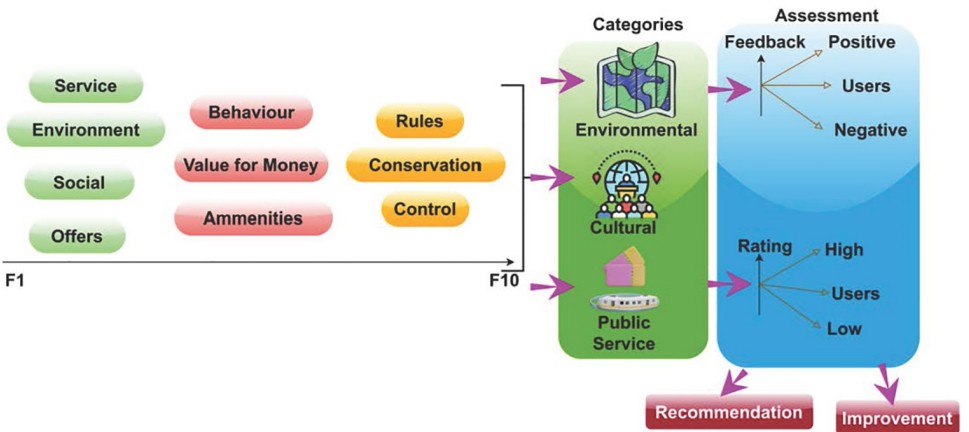

**Fig 1. F1 to F10 assessment data representation.**

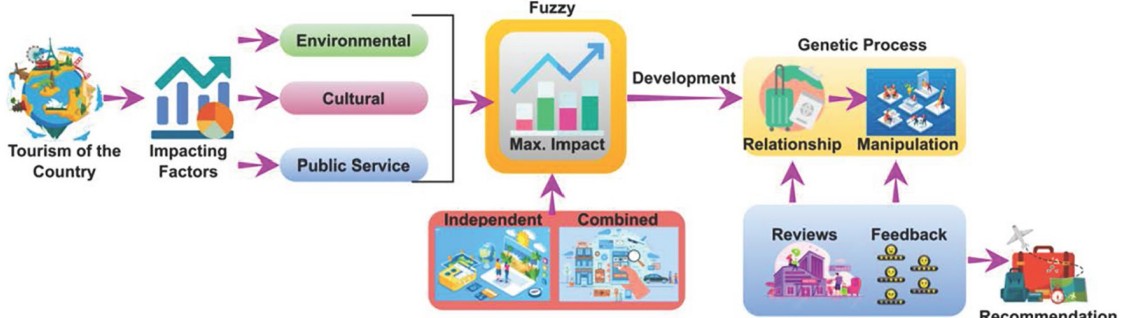

**Fig 2. Schematic of the proposed IDA method with the cumulative factors.**

algorithms that incorporate the impact factors. The impact factors are analyzed using feedback and ratings through independent and combined relationships and manipulations (refer to Fig 1). Intelligent tourism promotion is developed using statistical and accumulated data to improve recommendations and feedback. Here, the recommendation is processed to improve tourism in the countries. The impact factor is used to estimate the tourism guide that deploys the environmental, cultural, and public services that incorporate the cumulative factors. The impact factor is monitored for every set of processes and provides the relationship and manipulation. In this work, fuzzy theory is used to develop the impact factor. Here, the analysis is used to provide the best recommendations for tourism. The proposed assessment method is shown in Fig 2 below.

The scope of this paper is to improve the development process for this intelligent development assessment method (IDAM) using cumulative factors (CFs). The impact factor indicates a better relationship between the independent and combined features. The fuzzy process enhances the maximum possible impact factor through an independent evaluation, which is reviewed. Here, the output is integrated with the development factor, including the fuzzy and genetic processes, to better identify impact factor values. Based on the reviews and feedback, decisions are made to enhance the genetic process. The following equation calculates the cumulative environmental, cultural, and public service factors.

$$\partial = \begin{cases} (\rho * s_m) + \left( \dfrac{\sum_{f'} (u_p + r_t)}{v_e} \right) + e' * \varepsilon, \forall Environmental \\[3mm] \prod_{\varepsilon}^{v_e} (f' + s_m) * \left[ \dfrac{(v_e/d')}{(k_c + r_t)} \right] * (i_v + a_c) - u_p, \forall\, Cultural \\[3mm] \left[ (s_m + v_e) + \left( d' - u_p \right) \right] * \sum \left( k_c + \dfrac{\varepsilon}{f'} \right) + b', \forall\, Public\ service \end{cases} \quad (1)$$

The impact factor is found for the values and deploys the better tourism analysis to the country. The impact factor is recognized for the different sets of processing and estimates of the hybrid system. The finding is represented as $\partial$, and the current and previous values are described as $u_p$ and $r_t$, respectively. The development of this tourism process is represented as $\varepsilon$, and the feature value here is represented as $f$. The impact factor for this equation is denoted as $v_e$, and that for tourism is denoted as $s_m$. Periodic checking is necessary to monitor the condition, and it is represented as $k_c$. The decision is made whether the impact factor is provided to the particular tourism or not, and it is denoted as $d'$. The independent variable is represented as $\rho$. The visits and interactions that take place in cultural tourism are represented as $i_v$

and $a_c$, respectively. Environmental tourism includes $e'$, which states that the guide $b'$ symbolizes public infrastructure. The finding is achieved by deploying the impact factor and providing better decision-making. The cumulative factor is indicated for better decision-making for the environmental, cultural, and public service impact factors. The cumulative factor is derived from the hybrid system and provides the independent and combined factors. Here, the first case indicates environmental, the second is cultural, and the third is public service. All three factors result in the value of the cumulative factor. These factors in tourism are used to state the previous checklist and follow the current status. Here, the tourism guide is used to derive the places and explanations regarding the environmental factor, and it is represented as $\left( \frac{\sum_{f'}(u_p + r_t)}{v_e} \right) + e'$. Cultural involves the visits and interactions among tourism and the guide, and it is formulated as $*$. Finally, public service includes public infrastructure such as schools, colleges, etc., and is represented as $\sum \left( k_c + \frac{\varepsilon}{f'} \right) + b'$. In this way, the impact factor is processed to attain the value for tourism development. After the values from these cumulative factors are obtained, fuzzy theory is used to determine the value and impact factor for the above equation.

## 3.2 Fuzzy process for maximum impact assessment

Fuzzy theory is the process of assigning the numerical value of the system and making decisions. Here, the membership function defines the input for the fuzzy sets, which are assigned values. It is the boundary of information that is not clear to process and is defined as fuzzy theory. The main task of this theory is to state whether it relies on a particular boundary or not. The numerical value is derived as [0,1], where either the value relies on 0 or 1; based on this constraint, the fuzzy set is performed in the different states of the boundary. The proposed work uses fuzzy theory for maximum impact. Thus, the following equation is used to determine the value and impact factor to maximize only the impact factor.

$$\varrho = \left[ (\partial + v_e) + \left( f' * \frac{\rho}{(i_v + a_c)} \right) \right] * (e' + \varepsilon) - \left[ \left( {r_t}/{b'} + d' \right) + \psi \right] - d' \qquad (2)$$

This recognition can improve the maximum impact factor and aid in the development process. Here, the independent process evaluates the environment and cultural and public services. The recognition is described as $\varrho$, where $\psi$ is a recommendation. The independent and combined factors are processed for the maximum impact factor. The visits and interactions are derived from cultural factors, and the value intended is estimated from the cumulative factors. In this recognition step, the recommendation is made from the fuzzy operation in the numerical value. The unshaped boundary is found using a fuzzy operation. Here, recognition is achieved by deploying the development process. The independent and combined factor assessments using fuzzy theory are presented in Fig 3.

The $\partial$ analysis is performed for the independent factors $(s_m,1)$, $(s_m,1)$, and $(s_m,1) \forall \varepsilon, u_p$. The independent factors are analyzed as $b'0$ (low) or 1 (high), from which the peak $\partial$ is identified. If $(s_m, 0|1)$ is the output, then the combined assessment (i.e., $Q$) is put forward for high and low $\in$. It is obvious that both are capable of achieving 0 or 1 focused on development and recommendations for the travelers and services provided (Fig 3). Fig 4 presents the average $Q$ observed for the independent and combined factors for the service and tourists.

The left and right sides of Fig 4 present the independent and combined $Q$ values assessed for F1 to F9 under the I, II, and III categories. The above values are estimated for features $\in 1 < features = 0$ (i.e., if 5 or more features are 0, then the output is 0. In contrast, the right side of the image illustrates the relationship-based output for which 0 and 1 are rare phenomena

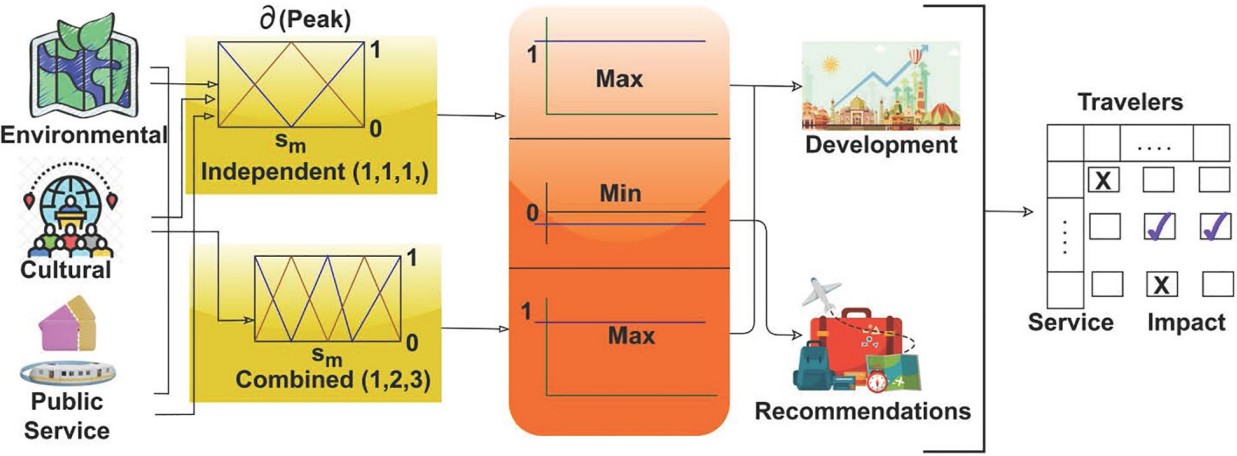

**Fig 3. Independent and combined factor assessment using fuzzy theory.**

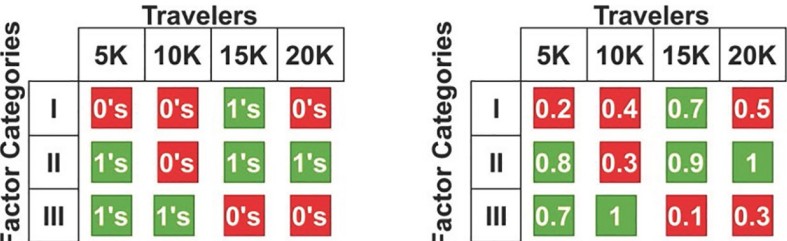

**Fig 4. Avg. Q observed for independent and combined factors.**

because either of the categories achieves a relationship $\neq 0$. This assessment is revisited later based on relationships and manipulations estimated using the genetic process. The independent process is termed for features that include visiting and interaction, and it is represented as $\left(f' * \frac{\rho}{(i_v + a_c)}\right)$. The recommendation is performed for decision-making and provides the current and previous state of matching for the efficient tour guide. This value is attained by using a decision-making approach to vary the guide information and places to visit. Based on this process, recognition is carried out to enhance the development method of fuzzy set theory. Here, the independent state derives the maximum impact factor and provides recommendations. The following equation is used to assign the degree of the membership function.

$$\chi = \{(\rho, b_i(\rho)) | \rho \epsilon (v_i, c_l, p_r)\} \tag{3}$$

The above Eq (3) derives the impact factor from the membership function. Here, the membership function is represented as $b_i$, where the assumed value ranges from [0,1]. The environmental, cultural, and public service factors are described as $v_i$, $c_l$, and $p_r$, respectively. In this membership function, the three cumulative factors are involved in obtaining universal information for tourism. The assignment of the membership function is symbolized as $\chi$; at this boundary, it is detected for the tourism recommendation. The fuzzy sets are constructed by mapping the current and previous states. Under these conditions, features are extracted from tourism and used to determine the degree of membership. After this method, the sigmoid

function is formulated in the following equation by using the membership function.

$$\sigma(\chi) = \frac{1}{1 + (v_i * c_i * p_r)} + \sum\nolimits_{f'}^{b_i} (\mu + n^{-\chi}) \tag{4}$$

Eq (4) is the sigmoid function, which derives the input from the membership function. The sigmoid function is described as $\sigma$, and the maximum impact is identified using this sigmoid function and is denoted as $\mu$, where $n$ is the Eucler function. The sigmoid function is used to fuzzify the numerical value that ranges from [0,1] and provides the result as a better recommendation. The sigmoid function identifies boundary information, including the environmental, cultural, and public service boundary information. Thus, the sigmoid function is computed for boundary detection in the tourism department. Then, this maximum impact is defined by checking the interval of time. The following equation checks the maximum impact for the tourism department.

$$k_c = \prod\nolimits_{e'}^{d'} (i_v + b') * \left(\frac{\sigma + \rho}{b_i / \mu}\right) + \sum\nolimits_{\chi} [(\psi * b_i) + (v_i + c_l + p_r)] - v_e \tag{5}$$

The maximum impact is observed by checking the value of the cumulative factor, which is represented in the above equation. Here, the guide is provided for visiting tourism, which includes the public infrastructure. The decision is made based on the interaction of tourism with the guide, and reviews and feedback are given. The recommendation is based on the decision made by the tourist. The fuzzy set is used for the independent and combined sets, and the impact is observed. The maximum impact is examined in this process, the recommendation is monitored, and it is represented as $\sum_{\chi} [(\psi * b_i) + (v_i + c_1 + p_r)]$.

In the above equation, the input is extracted from the fuzzy membership function, and the sigmoid function is used to fix the value of the boundary. Based on boundary detection, tourism is identified in the relationship between tourism and guides. The public service guide provides the infrastructure for tourists. In this manner, the guide provides recommendations for every cumulative factor. In this approach, the sigmoid function is used to rely on the values in the boundaries. Thus, checking takes place for every set of fuzzy processes. This method uses the following equations to define the independent and combined factors for impact improvement:

$$\rho = \left(\varrho * \frac{b_i + P}{k_c}\right) + \prod\nolimits_{\tau}^{p_r} (f' + v_e) - [s_m + (\varepsilon * \mu)] - \sigma \tag{6}$$

$$\nabla = \left(u_p + i_v\right) * \left(\frac{\mu * a_c / e' + r_t}{\varepsilon * \chi}\right) + Y * b_i \tag{7}$$

Eqs (6) and (7) describe the independent and combined factors used to improve the impact of the cumulative factor. Tourism recommendations are improved based on interaction and visiting. An independent study was conducted on the impact factors, examining the features and development of the tourism process. The environmental, cultural, and public service impact factors are obtained by deploying the maximum impact factor, which is represented as $[s_m + (\varepsilon * \mu)]$. Here, the sigmoid function is used in the fuzzy set for the membership function. The interaction and guide are provided for tourism to improve the impact factor. The previous and current state is used for the recommendation mapping. The fuzzy set for the independent and combined factors shows the maximum impact. The maximum and minimum impact

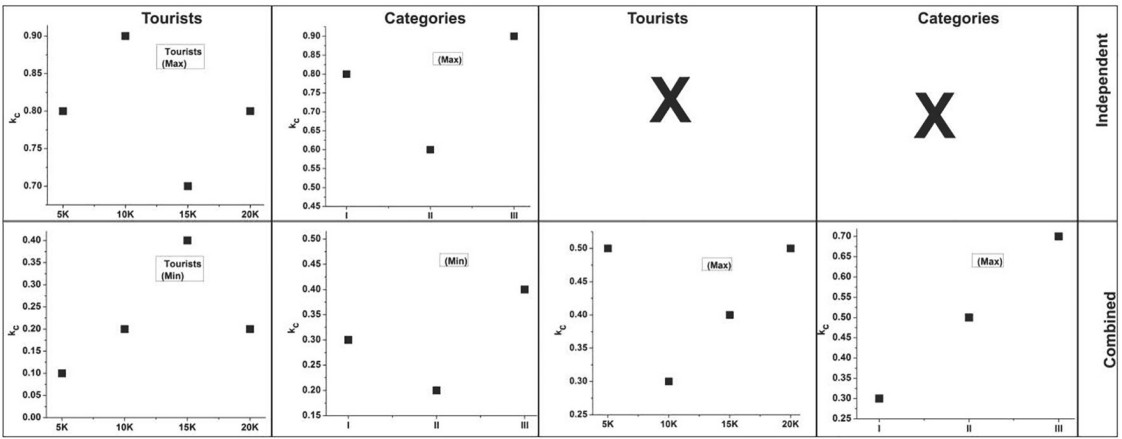

**Fig 5. Minimum and maximum impact factors for tourists.**

factors for tourists and the factors related to the membership degree function are analyzed in Fig 5.

In Fig 5, the $k_c$ analyses for tourists (min, max|| independent, combined) and categories (min, max|| independent||Combined) are presented. It is obvious that $k_c$ is 0 for min $\in$ tourists and is a category for independent factor assessments. However, due to $\rho$, $\nabla$, the combined methods are used to achieve a minimum and maximum output between 0 and 1. Therefore, the combined outputs are reliable for providing high recommendations for maximum impact. In this case, the independent factors other than $s_m$ (1,2,3) are utilized for $s_m(1)$. Therefore, the $k_c$ variants are distinguishable for combined use other than independent use. Therefore, the relationships and manipulation factors are assessed using a genetic algorithm. In Eq (7), the interaction is obtained for the current and previous states, which is derived from the fuzzy set and is represented as $Y$. The fuzzy set is examined to improve the maximum impact of tourism development. In this case, the public infrastructure is obtained for the features, provides the combined factors, and is symbolized as $\nabla$. Thus, the combined and independent factors are processed separately to enhance the impact. In this step, the value of the cumulative factor decreases, so the above Eqs (6) and (7) are integrated. The following equation is used to formulate the integration below.

$$\varepsilon = \prod_{f'}(\rho + \nabla)*\left(\mu*\frac{(\varsigma + \varrho)}{Y}\right) + \chi(v_e)*s_m - \psi \tag{8}$$

The development is obtained to improve the maximum impact by using fuzzy sets. Here, there are independent and combined factors. The input is the fuzzy set, and based on this, tourism is evaluated. The recommendation is based on this study's previous interaction with the guide regarding tourism. The development factor is used to determine the impact of the cumulative values. Here, recognition is used to examine the independent and combined factors. The maximum impact factor is achieved from this integration process, after which the genetic process is carried out to determine the relationship, and manipulation is performed to achieve better reviews and feedback.

### 3.3 Genetic process for relationship and manipulation assessment

The genetic process is used to obtain a high-quality solution using optimization methods. It is classified into four subtopics: inheritance, mutation, selection, and recombination. Here, the task is performed on a generated fuzzy set that deploys the reviews and feedback. This relationship and manipulation are performed through a genetic process that involves mutation and recombination (i.e., crossover). The following equations are formulated for mutation and crossover.

$$m_n = \left[ f'(v_e) * \left( u_p + d' \right) \right] + \sum_{b_i} (\varrho + s_m) \tag{9}$$

$$c_r = \frac{1}{v_i + c_l + p_r} * \varepsilon + \prod_{\chi} (\varsigma + \Omega) - Y \tag{10}$$

In the above equations, mutation and crossover are obtained, and they are represented as $m_n$ and $c_r$. Thus, the features are acquired and processed in this genetic process, and the identification method is derived. Identification is carried out to enhance the reviews and feedback based on this tourism scenario, which is improved through the use of a genetic process. The best fitness value mutation is obtained. This method evaluates different types of tourism and provides feedback regarding the request. This previous state is estimated for the decision-making approach. Eq (10) is obtained by integrating the relationship and manipulation and is represented as $\varsigma$ and $\Omega$. Thus, the relationships between the cumulative factors and the obtained values are determined. The manipulation is performed by modifying the impact factor and enhancing the recommendation. Thus, mutation and crossover occur during this genetic process. The following equation is used to identify the reviews from the relationship.

$$\tau(\varsigma) = (\eta + s_m) * \left[ \frac{\left( {Y+P}/{\rho + \nabla} \right)}{(\varrho + k_c) * \varsigma} \right] + \left[ (m_n * c_r) + (\psi * \varepsilon) \right] + \Lambda \tag{11}$$

The identification of the reviews from the relationship utilizes both the independent and combined factors. The identification is represented as $\tau$, and the genetic process is described as $\Lambda$. The process enhances the recommendation for tourists accessing services through the cumulative factor. Based on this feedback, reviews are improved in this system. For every genetic process, identification is carried out for the reviews, which are derived from the relationship factor of the mutation method. After this identification of reviews from the mutation method, the manipulation is performed using the equation below.

$$\tau(\Omega) = \left[ \left( \Lambda + \frac{\mu}{\eta} \right) * \varrho \right] + (\rho * \nabla) - \psi \tag{12}$$

The identification is obtained for the reviews based on the manipulation process modified from the cumulative factors. Here, the genetic process includes the relationship between the reviews and manipulation methods. These methods examine the proposed work's value and impact factor. The relationship-based genetic operations are illustrated in Fig 6.

The genetic algorithm's process for 3 categories, $m_n$ and $C_r$ is defined in Fig 6. This process relies on clubbing the maximum possibility for various reviews and feedback. The $c_r$ for multiple categories are split for various reviews (high/low) and types of feedback (good/bad) with the combined factor analysis. This process identifies the relationship between the impact factors F1 and F10 from which the manipulation (without repetition) is extracted. Fig 7 presents the feedback and review-based $Q$ results (from Fig 4).

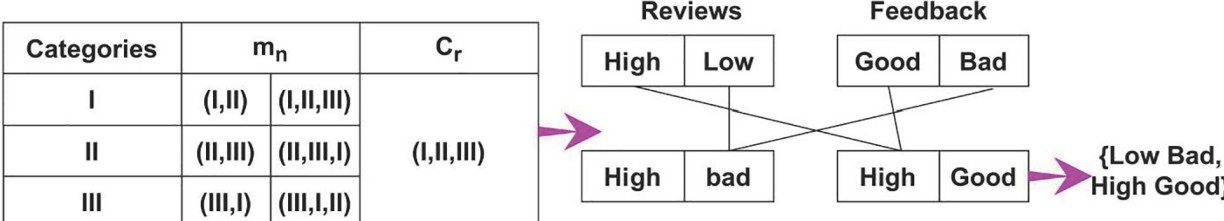

**Fig 6. Relationship-based genetic operations.**

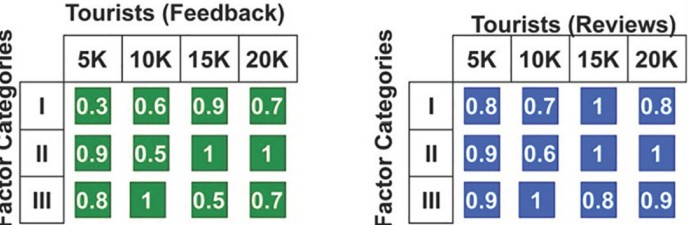

**Fig 7. Q (*Avg*) observed based on the reviews and feedback.**

In Fig 7, $Q$ is revisited using $m_n$ and $c_r$ with the reviews and feedback operations. The proximal case of a review obtained from tourists is optimal for deciding the impact factor from which the recommendations are given. In this process, the $Q$ for the independent factor analysis is prevented because it increases the chances of obtaining a 1. Therefore, the combined factors are used to identify the least possible factor for tourist recommendations with better attractions. This analysis thus increases the chances of making development recommendations. The recommendation process is performed here by deploying value and impact factor recognition. This is achieved by deploying the impact factor and enhancing the process. The manipulation is performed to find the best reviews in this genetic process. This manipulation process includes independent and combined factors that enhance the recommended method. This feedback is performed using the equation below.

$$\xi = \frac{1}{\psi * \eta} + \sum_{Y}^{v_e} \left( d' + P \right) * k_c \tag{13}$$

In the above equation, feedback is given to obtain the recommendation process that determines the impact factor. The feedback is represented as $\xi$, where the impact factor is improved. The evaluation process is carried out considering the various impact factors and relationships. The relationship and manipulation are performed in the genetic process, and feedback is provided based on this review. The feedback is based on the cumulative factor and provides a better recommendation from the fuzzy set. This proposed work focuses on the genetic process of development and tourism recommendations. Thus, the scope of this work is satisfactory.

## 4. Result analysis

This discussion is presented as a cumulative analysis using the recommendation ratio, development rate, factor detection, review-based assessment, and assessment time. According to the data considered (Ningbo Bureau of Statistics, 2023), nearly 10 locations see 710K visitors

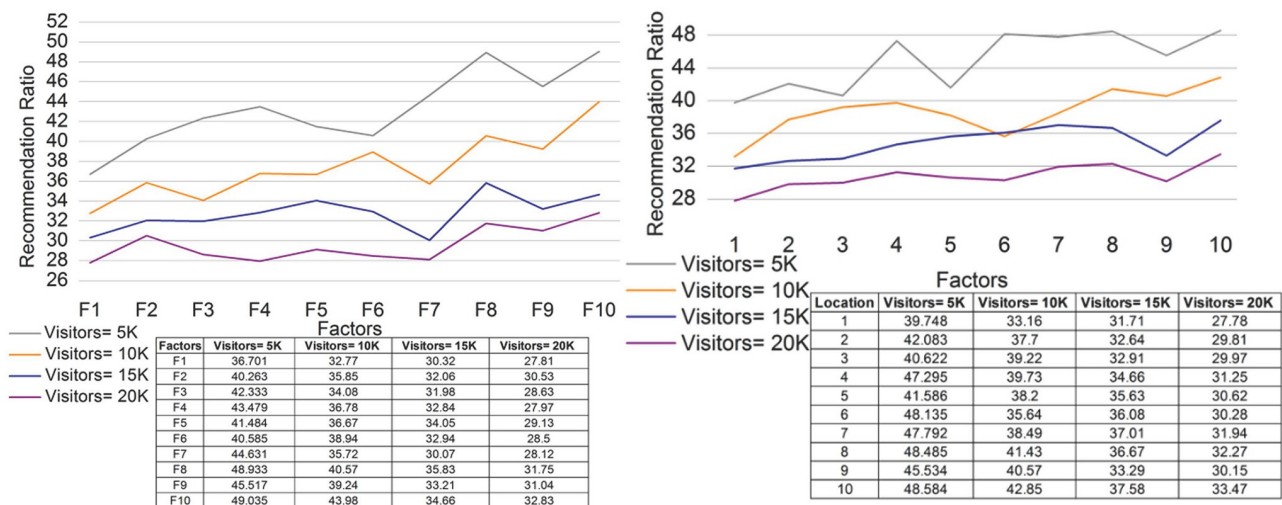

**Fig 8. Discussion of the recommendation ratio for (a) factors and (b) locations.**

per year. Therefore, 10 impact features (as shown in Fig 1) and 10 locations are considered variables in the data input. The number of visitors in this analysis is 5K, 10K, 15K, and 20K.

## 4.1 Recommendation ratio

The recommendation ratio for the proposed work increases for various factors and locations. The recommendations in Fig 8(a) and 8(b) are based on the fuzzy set that includes the maximum impact. Here, the impact factor and membership function are derived for the independent and combined factors. In this approach, the relationship is examined using the genetic process. Eq (2) is used to recognize the value and impact factors, and it is represented as $\left[\left({r_t}/{b'+d'}\right) + \psi\right]$. The recommendation is improved in this work for various values of the cumulative factor. The membership function is defined using the fuzzy set and provides feedback and reviews. The feedback and reviews are evaluated for the manipulation of services. The cumulative factor is used for the value assignment, which is then used to obtain the value for tourism. A tourism recommendation is provided for the locations. The maximum impact is enhanced in this work by improving the recommendation. If the recommendation is improved, the genetics and reviews are processed and added to the guide to visit the particular place.

## 4.2 Development rate

In Fig 9(a) and 9(b), the development rate is enhanced for the different factors and locations. Here, the development rate is enhanced in this approach, and the value-based assignment is provided. The relationships and manipulations are processed in the fuzzy set and used to enhance the impact factor. The maximum impact factor is enhanced and provides the independent and combined factors, and it is represented as $\left(\mu * \frac{(\varsigma+\varrho)}{Y}\right)$. The recognition and relationships are estimated based on the maximum impact. The development is based on the genetic process, including the previous computations. The computation relies on a fuzzy set that includes feedback and reviews. Here, the development rate is improved, and the reviews and feedback for the cumulative factor are shown. The cumulative factors are observed by the values of the relationship and manipulation. Both processes include recognition and feedback

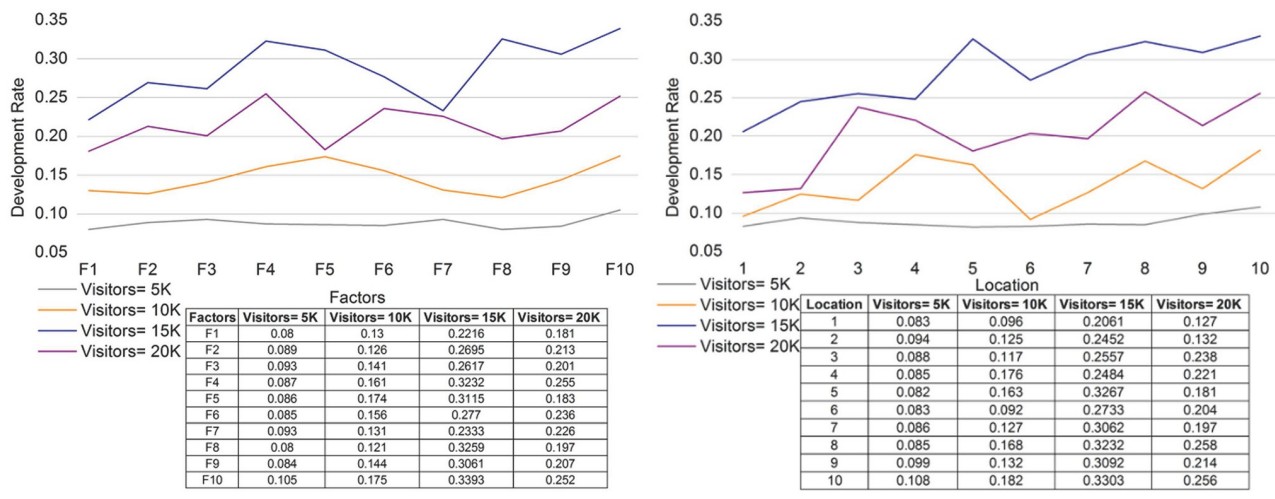

**Fig 9. Discussion of the development rate for (a) factors and (b) locations.**

for better reliability. Thus, the development is obtained using Eq (8), where the input is the fuzzy set, and Eqs (7) and (8) are integrated and include the independent and combined factors.

## 4.3 Factor detection

Factor detection is improved based on the tourism location and provides the cumulative factor for the reviews and relationships. Here, the genetic process is involved in this approach, and the relationship and manipulation are estimated. In Fig 10, the detection factor of recommendations and reviews shows the integration of the relationships among features. Matching is performed with the current and previous sets and provides the interaction for tourism, which

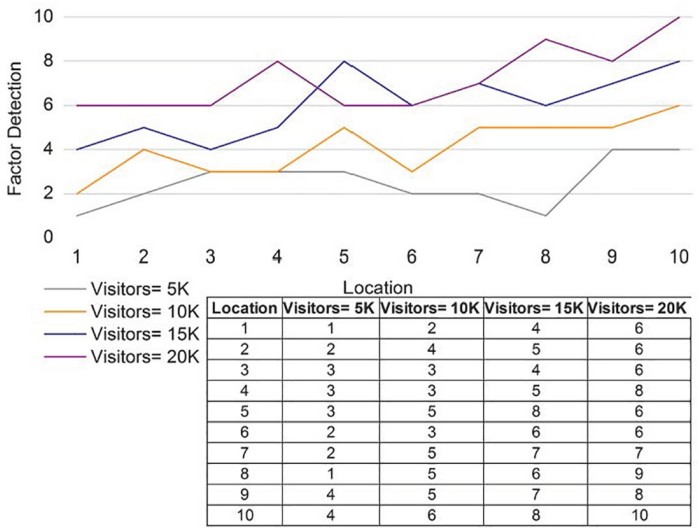

**Fig 10. Discussion of factor detection for the different locations.**

is described as $[(m_n * c_r) + (\psi * \varepsilon)]$. Development is achieved by deploying the independent and combined factors, including the tourism membership function. The cumulative factor provides the services' sigmoid function, including the identifying features. The feature-based process is performed for tourism and derives the genetic process, and it is symbolized as $\left[\dfrac{\left(Y+P/_{\rho+\nabla}\right)}{(\varrho+k_c)*\varsigma}\right]$. The integration of independent and combined factors is performed through the genetic process and reveals reliable detection factors for tourism based on location. In this case, the location is detected accurately and delivers a better tourist experience.

## 4.4 Review-based assessment

The review-based assessment is improved for various locations and factors, and the value feedback is evaluated. The values of the factors are examined to improve the maximum impact. If the impact factor is improved in the cumulative part, the review-based assessment is enhanced. In Eq (9), the genetic process that deploys the mutation is examined, and crossover is performed. The genetic process includes reviews and maximum impact, and it is formulated as $\left[\left(\Lambda + \frac{\mu}{\eta}\right)*\varrho\right]$. This is achieved by identifying the reviews established using the relationships and manipulation. Here, both constraints are used to derive the feature recommendation for tourism. Tourists select a particular location to visit, and this impact factor is assigned to improve the membership function. The genetic process is carried out using various approaches, including the fuzzy set for the membership function. Recommendations are made for various tourism locations. In this case, reviews are given to the tourist to select a location to visit, which enhances the output [Fig 11(a) and 11(b)].

## 4.5 Assessment time

In Fig 12(a) and 12(b), the assessment time decreases for the various locations and factors considered in the proposed work. Here, independent and combined factors are involved to maximize the impact and improve the computation. The genetic process is performed for feedback, reviews, and decision-making, and it is represented as $[f'\,(v_e) * (up + d')]$. The recognition is performed for the values and factors in this method. The independent and combined values

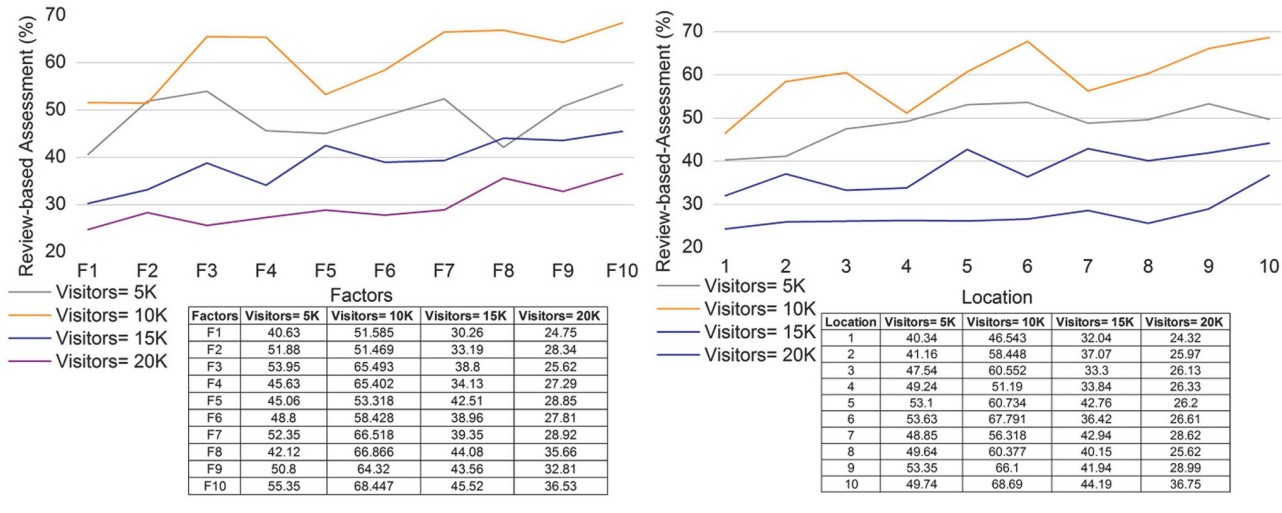

**Fig 11. Review-based assessment for (a) factors and (b) locations.**

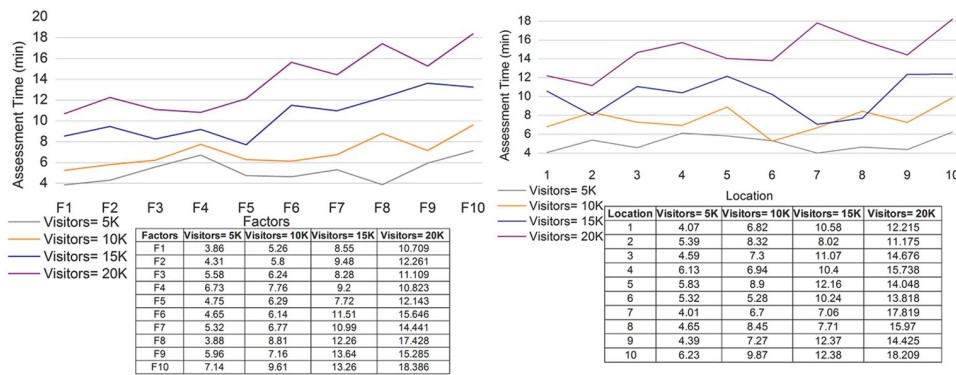

**Fig 12. Assessment time for (a) factors and (b) locations.**

are used to develop the reviews and feedback. Here, the assessment time for the proposed work decreases and shows the efficient development of the genetic process. In this case, mutation and crossover are included in the environmental, cultural, and public service factors. Thus, the assessment time for the proposed work shows that the most reliable method for the hybrid system includes fuzzy and genetic processes. The sigmoid function is related to the degree of membership value and provides the impact factor. Periodic checking is performed for recommendations before reviews and feedback are given, decreasing the assessment time.

## 5. Conclusion

This article discussed the processes and performance of the intelligent development assessment method for improving tourism using its allied services. Environmental, cultural, and public service interactions are considered impact factors for deciding on development. These impact factors are independently and cumulatively handled using fuzzy and genetic operations for better recommendations. In the fuzzy process, the maximum impact on tourism development is assessed. Using the fuzzy output, the relationship between the factors is constructed and manipulated to identify the maximum positive impact. The genetic outputs through crossover and mutation identify different outputs of cumulative factor connectivity. Therefore, the proposed method identifies the maximum possible recommendations based on the reviews and feedback provided by the tourists from their previous experiences. Overall, the genetic process provides the optimal recommendations for attracting tourists. The system's efficiency was assessed using the suggestion ratio, which ensured a 48.58% success rate, a development rate of 0.105%, a 4-factor detection rate, and a review-based assessment rate of 55.5% for a sample size of 5,000 visitors.

One potential constraint may be data availability and quality, as the system depends on complete data inputs to conduct precise analysis. Subsequent investigations may delve into the integration of supplementary data sources and the enhancement of data preparation methodologies. Furthermore, conducting a comprehensive assessment of the system's performance at various tourism sites is recommended to improve its applicability and resilience.

## Author Contributions

**Investigation:** Jinxia Lou.

**Methodology:** Jinxia Lou.

**Resources:** Jinxia Lou.

**Supervision:** Jinxia Lou.

**Writing – original draft:** Jinxia Lou.

**Writing – review & editing:** Jinxia Lou.

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
