## [Decision Letter · Decision Letter 0]

27 Mar 2024

PONE-D-24-08338Analyzing Intelligent Tourism Development and public services based on a Fuzzy Genetic Hybrid System to Promote Environmental and cultural valuesPLOS ONE

Dear Dr. Lou,

Thank you for submitting your manuscript to PLOS ONE. After careful consideration, we feel that it has merit but does not fully meet PLOS ONE’s publication criteria as it currently stands. Therefore, we invite you to submit a revised version of the manuscript that addresses the points raised during the review process.

Please carefully check the Reviewers’ comments and improve the manuscript. Comments from PLOS Editorial Office: We note that one or more reviewers has recommended that you cite specific previously published works. As always, we recommend that you please review and evaluate the requested works to determine whether they are relevant and should be cited. It is not a requirement to cite these works. We appreciate your attention to this request.

We look forward to receiving your revised manuscript.

Kind regards,

Agnieszka Konys, Ph.D.

Academic Editor

PLOS ONE

Journal Requirements:

3. In the online submission form you indicate that your data is not available for proprietary reasons and have provided a contact point for accessing this data. Please note that your current contact point is a co-author on this manuscript. According to our Data Policy, the contact point must not be an author on the manuscript and must be an institutional contact, ideally not an individual. Please revise your data statement to a non-author institutional point of contact, such as a data access or ethics committee, and send this to us via return email. Please also include contact information for the third party organization, and please include the full citation of where the data can be found.

4. PLOS requires an ORCID iD for the corresponding author in Editorial Manager on papers submitted after December 6th, 2016. Please ensure that you have an ORCID iD and that it is validated in Editorial Manager. To do this, go to ‘Update my Information’ (in the upper left-hand corner of the main menu), and click on the Fetch/Validate link next to the ORCID field. This will take you to the ORCID site and allow you to create a new iD or authenticate a pre-existing iD in Editorial Manager. Please see the following video for instructions on linking an ORCID iD to your Editorial Manager account: https://www.youtube.com/watch?v=_xcclfuvtxQ.

5. Please be informed that funding information should not appear in the Acknowledgments section or other areas of your manuscript. We will only publish funding information present in the Funding Statement section of the online submission form. Please remove any funding-related text from the manuscript.

Reviewers' comments:

Reviewer's Responses to Questions

**Comments to the Author**

1. Is the manuscript technically sound, and do the data support the conclusions?

Reviewer #1: Partly

Reviewer #2: Yes

2. Has the statistical analysis been performed appropriately and rigorously? 

Reviewer #1: N/A

Reviewer #2: Yes

3. Have the authors made all data underlying the findings in their manuscript fully available?

Reviewer #1: No

Reviewer #2: Yes

4. Is the manuscript presented in an intelligible fashion and written in standard English?

Reviewer #1: No

Reviewer #2: No

5. Review Comments to the Author

Reviewer #1: Dear author,

I appreciate the novelty of the work. It has many merits and is able to provide an alternative approach, as you rightly mentioned. However, at the current phase, I have observed some lapses in your manuscript. Please find my observations below:

1. The Abstract is unclear, especially regarding the last part and how it concludes and is useful as an alternative approach. Adding a conclusion and recommendation line in the abstract would benefit readers' understanding.

2. The Background section looks incomplete; it misses the aim and scope of the study at the end. The required information seems to be written below Figure 2, but it should be included in the Background section. I suggest authors check published papers in PLOS One and try to replicate the style for uniformity.

3. The Literature Review is weakly developed. I suggest authors write the Literature Review section in paragraphs rather than in the current format. Here too, please refer to earlier published papers in PLOS One for guidance.

4. In Kong et al. [32], the authors wrote, "Then I will analyze them based on the traditional theory of the tourism industry." This sentence is not connected anywhere. Please ensure all paragraphs and sentences are well-written and connected.

5. The Methods section is not properly developed for this study. I have found it in various places but not in the specific point. I, therefore, suggest authors write it after the Literature Review section and then follow their logical argument. Please refer to published papers in PLOS One for guidance. It will help to develop their argument as well as maintain the standard of the journal.

6. I suggest combining Section 3, Section 4, and Section 5 under a single heading. It could be titled "Purpose Model/Method" as it includes mathematical and other technical aspects showing how it can go vis-a-vis. This type of heading leaves a greater impression on the readers.

7. Instead of 'Discussion', consider using different headings such as "Result Analysis" or "Cumulative Analysis."

8. Replace the heading "Summary" with "Conclusion" and ensure it is written properly.

9. Apart from Background and Literature Review, the paper is not cited properly. It needs to be developed as a research paper. Please adhere to the standards set by PLOS One for writing.

10. One important observation is that this manuscript has not been well-cited and does not use previous papers as references. References and citations are limited only to the literature review, and thereafter, all writing is done without any scientific support. It needs to cite and compare other studies and findings with the proposed model presented in this paper and other sections of the paper.

Since this paper proposes new insights which could take a bigger shape in the future, it is worth publishing. However, the development of the current form is very weak and requires professional writing. Following PLOS One guidelines could be beneficial.

Reviewer #2: The first heading of the paper should be INTRODUCTION rather than Background.

The term in the first paragraph of the paper "social Culture legacy" should be "social culture legacy" or "Social Culture Legacy".

The review of literature is in chronological order. It should be systematic and logical. This section requires overall revision. The section should conclude to the research gap addressed in this research. Furthermore, the literature should be enriched with the insights from the most recent studies on the role of tourism and its impact on socio-economic life and environment. For instance,

https://doi.org/10.1177/13548166211000480

https://link.springer.com/article/10.1007/s11356-023-28377-0

https://doi.org/10.3390/admsci13080172

https://link.springer.com/referenceworkentry/10.1007/978-3-030-48652-5_110

There is no comprehensive discussion on the results.

The Summary section needs revision.

First of all the authors should discuss the findings of the study.

The most relevant policy implications followed by the limitations and future directions of research should be spelled out.

6. PLOS authors have the option to publish the peer review history of their article (what does this mean?). If published, this will include your full peer review and any attached files.

Reviewer #1: No

Reviewer #2: No

---

## [Author Response · Author response to Decision Letter 0]

10 Apr 2024

Reviewer #1:

I appreciate the novelty of the work. It has many merits and is able to provide an alternative approach, as you rightly mentioned. However, at the current phase, I have observed some lapses in your manuscript. Please find my observations below:

1. The Abstract is unclear, especially regarding the last part and how it concludes and is useful as an alternative approach. Adding a conclusion and recommendation line in the abstract would benefit readers' understanding.

Ans:

The efficiency of the system evaluated using the recommendation ration in which system ensures 48.58%, 0.105% development rate, 4 factor detection rate, 55.5% review based assessment for 5k visitors.

2. The Background section looks incomplete; it misses the aim and scope of the study at the end. The required information seems to be written below Figure 2, but it should be included in the Background section. I suggest authors check published papers in PLOS One and try to replicate the style for uniformity.

Ans:

The objective of this system is to conduct an analysis and facilitate the advancement of intelligent tourist development, taking into account both environmental and cultural values. The assessment of public services and infrastructure connected to tourism is conducted using a fuzzy genetic hybrid technique. The technology enhances decision-making processes for sustainable tourism practices by integrating fuzzy logic and genetic algorithms. In essence, its objective is to achieve a harmonious equilibrium between economic expansion and the conservation of natural resources and cultural legacy. In figure 2 below describe that what is going to discussed in the particular section. The PLOS based article is checked accordingly. 

3. The Literature Review is weakly developed. I suggest authors write the Literature Review section in paragraphs rather than in the current format. Here too, please refer to earlier published papers in PLOS One for guidance.

Ans:

Literature survey is changed into the paragraph according to the comment.

4. In Kong et al. [32], the authors wrote, "Then I will analyze them based on the traditional theory of the tourism industry." This sentence is not connected anywhere. Please ensure all paragraphs and sentences are well-written and connected.

Ans:

Reference [32] explanation changed according to the comment.

[32] presents a novel real-time processing system and Internet of Things (IoT) application designed to facilitate the development of cultural tourism. The utilization of Internet of Things (IoT) sensors and big data analytics is employed to effectively monitor and manage tourism resources. The system conducts real-time data processing from many sources in order to generate valuable insights that can inform decision-making processes. Furthermore, it integrates sophisticated algorithms to optimize the allocation of resources and improve the visitor experience, all while fostering the preservation of cultural assets.

5. The Methods section is not properly developed for this study. I have found it in various places but not in the specific point. I, therefore, suggest authors write it after the Literature Review section and then follow their logical argument. Please refer to published papers in PLOS One for guidance. It will help to develop their argument as well as maintain the standard of the journal.

Ans:

The methodology section is checked according to the comment. The additional contents are included for improving the manuscript efficiency.

The primary aim of this system is to promote the sustainable and rational development of tourism, while simultaneously ensuring the protection of environmental and cultural resources. The objective is to achieve a state of equilibrium that promotes both economic expansion via tourism and the conservation of natural resources, ecological systems, and cultural legacy. The system seeks to enhance decision-making processes pertaining to public services, infrastructure, and resource allocation in the tourism industry through the utilization of a fuzzy genetic hybrid approach. The proposed hybrid methodology integrates the advantageous aspects of fuzzy logic, which possesses the capability to effectively address uncertainties and imprecise data, with genetic algorithms, which provide efficient search and optimization techniques. The system aims to optimize the beneficial outcomes of tourism while mitigating any adverse effects on the environment and cultural values, employing a synergistic approach. In essence, the primary objective is to advance responsible tourism strategies that foster the enduring sustainability of destinations, so guaranteeing the ability of forthcoming generations to value and partake in their natural and cultural heritage.

6. I suggest combining Section 3, Section 4, and Section 5 under a single heading. It could be titled "Purpose Model/Method" as it includes mathematical and other technical aspects showing how it can go vis-a-vis. This type of heading leaves a greater impression on the readers.

Ans:

The section 3, section 4 and section 5 titles come under proposed model. 

7. Instead of 'Discussion', consider using different headings such as "Result Analysis" or "Cumulative Analysis."

Ans:

Heading changed accordingly. 

8. Replace the heading "Summary" with "Conclusion" and ensure it is written properly.

Ans:

Heading changed accordingly.

9. Apart from Background and Literature Review, the paper is not cited properly. It needs to be developed as a research paper. Please adhere to the standards set by PLOS One for writing.

Ans:

References are checked and cited properly according to the journal format and standards.

10. One important observation is that this manuscript has not been well-cited and does not use previous papers as references. References and citations are limited only to the literature review, and thereafter, all writing is done without any scientific support. It needs to cite and compare other studies and findings with the proposed model presented in this paper and other sections of the paper.

Ans:

Citations are checked and the proposed method having the contribution of this work. There is no need to citation for proposed method explanation which is the main contribution of the work. In addition, the citation and explanations are related to the journal standard.

Since this paper proposes new insights which could take a bigger shape in the future, it is worth publishing. However, the development of the current form is very weak and requires professional writing. Following PLOS One guidelines could be beneficial.

Reviewer #2: 

The first heading of the paper should be INTRODUCTION rather than Background.

Ans:

Heading changed according to the comment.

The term in the first paragraph of the paper "social Culture legacy" should be "social culture legacy" or "Social Culture Legacy".

Ans:

Changed into social culture legacy

The review of literature is in chronological order. It should be systematic and logical. This section requires overall revision. The section should conclude to the research gap addressed in this research. Furthermore, the literature should be enriched with the insights from the most recent studies on the role of tourism and its impact on socio-economic life and environment. For instance,

https://doi.org/10.1177/13548166211000480

https://link.springer.com/article/10.1007/s11356-023-28377-0

https://doi.org/10.3390/admsci13080172

https://link.springer.com/referenceworkentry/10.1007/978-3-030-48652-5_110

Ans:

These papers are included in the literature survey according to the comment.

Ahmad, N., & Ma, X. (2022) [33]examines the correlation between the growth of tourism and the occurrence of environmental contamination in various socioeconomic brackets of nations. This study employs panel data analysis to investigate the influence of various factors, including tourist arrivals, tourism receipts, and tourism investment, on air pollution, water pollution, and CO2 emissions. The results indicate a multifaceted relationship between the expansion of tourism and the deterioration of the environment, underscoring the necessity for the implementation of sustainable tourism laws and practices. The authors offer suggestions for mitigating the negative environmental impacts associated with the rise of tourism.

Ali, S et al. 2023 [34] examines the ecological consequences of global tourism, taking into account the influence of policy ambiguity, renewable energy, and service sector production. Drawing upon data obtained from prominent tourist locations, this study utilizes sophisticated econometric methodologies to examine the impact of these variables on carbon emissions and ecological footprints. The results of the study demonstrate complex interconnections, indicating that the presence of policy uncertainty poses obstacles to the adoption of sustainable practices. Conversely, the rise of renewable energy and the service sector have the potential to alleviate environmental deterioration resulting from tourism activities. The policy recommendations put out by the authors aim to enhance the promotion of environmental sustainability within the tourism industry.

García-Madurga, et al. 2023 [35] provides a comprehensive analysis of the evaluations pertaining to the utilization of Artificial Intelligence (AI) within the tourism sector. The process involves a methodical examination of available literature studies to consolidate the present status of AI implementation in different areas of the tourism industry. The research emphasizes the capacity of artificial intelligence (AI) technology to augment tourism experiences, operations, and decision-making procedures, while also revealing areas of deficiency and promising avenues for future investigation.

Bulchand-Gidumal, J. (2022) [36] investigates the effects of artificial intelligence (AI) on the travel, tourism, and hospitality industries. This study examines a range of artificial intelligence (AI) applications, including chatbots, recommendation systems, and predictive analytics, emphasizing its capacity to improve customer experiences and operational efficiency. The chapter additionally examines the obstacles and ethical implications associated with the implementation of artificial intelligence in certain sectors.

There is no comprehensive discussion on the results.

Ans:

The result and discussion described in section 4, the comparative analysis is conducted for various number of visitors. The related description described in section 4.

The Summary section needs revision.

Ans:

Summary revised according to the comment. The system's efficiency was assessed using the suggestion ratio, which ensured a 48.58% success rate, a development rate of 0.105%, a 4-factor detection rate, and a review-based assessment of 55.5% for a sample size of 5,000 visitors.

First of all the authors should discuss the findings of the study.

Ans:

Findings of the study is included in the end of the conclusion. The system's efficiency was assessed using the suggestion ratio, which ensured a 48.58% success rate, a development rate of 0.105%, a 4-factor detection rate, and a review-based assessment of 55.5% for a sample size of 5,000 visitors.

The most relevant policy implications followed by the limitations and future directions of research should be spelled out.

Ans:

One potential constraint may pertain to the availability and quality of data, as the system is dependent on complete data inputs in order to conduct precise analysis. Subsequent investigations may delve into the integration of supplementary data sources and the enhancement of data preparation methodologies. Furthermore, it is recommended to conduct a comprehensive assessment of the system's performance in various tourism sites in order to improve its applicability and resilience.

---

## [Editor Report · Decision Letter 1]

3 Jun 2024

PONE-D-24-08338R1Analyzing Intelligent Tourism Development and public services based on a Fuzzy Genetic Hybrid System to Promote Environmental and cultural valuesPLOS ONE

Dear Dr. Lou,

Thank you for submitting your manuscript to PLOS ONE. After careful consideration, we feel that it has satisfied our scientific requirements for publication.

However, our editorial team have significant concerns about the grammar, usage, and overall readability of the manuscript. PLOS ONE requires that published manuscripts use language which is 'clear, correct, and unambiguous', see our criteria for publication at https://journals.plos.org/plosone/s/criteria-for-publication#loc-5. We therefore request that you revise the text to fix the grammatical errors and improve the overall readability of the text.

We suggest you have a fluent English-language speaker thoroughly copyedit your manuscript for language usage, spelling, and grammar. If you do not know anyone who can do this, you may wish to consider employing a professional scientific editing service.

Whilst you may use any professional scientific editing service of your choice, PLOS has partnered with both American Journal Experts (AJE) and Editage to provide discounted services to PLOS authors. Both organizations have experience helping authors meet PLOS guidelines and can provide language editing, translation, manuscript formatting, and figure formatting to ensure your manuscript meets our submission guidelines. To take advantage of our partnership with AJE, visit the AJE website (https://www.aje.com/go/plos/) for a 15% discount off AJE services. To take advantage of our partnership with Editage, visit the Editage website (www.editage.com) and enter referral code PLOSEDIT for a 15% discount off Editage services. If the PLOS editorial team finds any language issues in text that either AJE or Editage has edited, the service provider will re-edit the text for free.

Please note that we will not be able to proceed with publication of your manuscript until the concerns above are addressed.

* A copy of your manuscript showing your changes by either highlighting them or using track changes (uploaded as a supporting information file)

* A clean copy of the edited manuscript (uploaded as the new manuscript file)

We look forward to receiving your revised manuscript.

Kind regards,

Joanna Tindall

Staff Editor

on behalf of

Agnieszka Konys, Ph.D.

Academic Editor

PLOS ONE
---

## [Author Response · Author response to Decision Letter 1]

17 Jun 2024

1. We therefore request that you revise the text to fix the grammatical errors and improve the overall readability of the text. We suggest you have a fluent English-language speaker thoroughly copyedit your manuscript for language usage, spelling, and grammar. If you do not know anyone who can do this, you may wish to consider employing a professional scientific editing service.

Ans: Proofreading is done. Grammatical errors have been rectified.

Ans: The retracted article has not been cited in this manuscript.

---

## [Editor Report · Decision Letter 2]

24 Jun 2024

Analyzing Intelligent Tourism Development and Public Services Based on a Fuzzy Genetic Hybrid System to Promote Environmental and Cultural Values

PONE-D-24-08338R2

Dear Dr. Lou,

We’re pleased to inform you that your manuscript has been judged scientifically suitable for publication and will be formally accepted for publication once it meets all outstanding technical requirements.

Kind regards,

Agnieszka Konys, Ph.D.

Academic Editor

PLOS ONE
---

## [Editor Report · Acceptance letter]

27 Jun 2024

PONE-D-24-08338R2 

PLOS ONE

Dear Dr. Lou, 

I'm pleased to inform you that your manuscript has been deemed suitable for publication in PLOS ONE. Congratulations! Your manuscript is now being handed over to our production team.

Kind regards, 

on behalf of

Dr. Agnieszka Konys 

Academic Editor

PLOS ONE